# A Mixture of Kaempferol-3-*O*-sambubioside and Kaempferol-3-*O*-sophoroside from *Malvaviscus arboreus* Prevents Ethanol-Induced Gastric Inflammation, Oxidative Stress, and Histologic Changes

**DOI:** 10.3390/plants11212951

**Published:** 2022-11-01

**Authors:** Yrvinn Campos-Vidal, Alejandro Zamilpa, Enrique Jiménez-Ferrer, Antonio Ruperto Jiménez-Aparicio, Brenda Hildeliza Camacho-Díaz, Gabriela Trejo-Tapia, Daniel Tapia-Maruri, Nayeli Monterrosas-Brisson, Maribel Herrera-Ruiz

**Affiliations:** 1Centro de Investigación Biomédica del Sur, Instituto Mexicano Del Seguro Social, Argentina # 1, Centro, Xochitepec 62790, Mexico; 2Centro de Desarrollo de Productos Bióticos, Instituto Politécnico Nacional, Col. San Isidro, Yautepec 62731, Mexico; 3Facultad de Ciencias Biológicas, Universidad Autónoma del Estado de Morelos, Cuernavaca 62209, Mexico

**Keywords:** *Malvaviscus arboreus*, kaempferol-3-O-sambubioside and kaempferol-3-O-sophoroside, cytokines, histologic analysis, catalase

## Abstract

*Malvaviscus arboreus* is used in traditional Mexican medicine to treat gastrointestinal diseases. Therefore, a mixture of Kaempferol-*O*-sambubioside and Kaempferol-*O*-sophoroside (MaSS) isolated from flowers of this species was tested as a preventive treatment on gastric lesions induced with ethanol in rats. MaSS was obtained by chromatographic methods and administered by oral pathway to male Sprague Dawley rats with ethanol-induced gastric lesions. Pretreatment with MaSS at doses of 30, 90, 120, and 180 mg/kg significantly prevents gastric lesions, inhibits the increment in relative stomach weight (%) in gastric IL-6, and also provokes an increment of IL-10 concentration and catalase activity. Finally, MaSS prevented edema in the mucosa and submucosa and diminished microscopic gastric lesions provoked by ethanol.

## 1. Introduction

Gastric ulcers (GU) are multifactorial and complex disorders, affecting an average of 14.5 million people worldwide annually, with a 4.08% mortality rate [1]. These gastrointestinal diseases have been related to an imbalance of protective factors (the integrity of the gastric mucosa) and aggressive factors (gastric acid secretion). It is characterized by necrosis, white blood cell infiltration, and an abrasive zone, among other features [2,3,4]. Despite the existence of the epithelial layer, which acts as a protective barrier in the gastrointestinal tract, ingested materials and pathogens can cause inflammation by activating the epithelium, neutrophils, and macrophages to produce inflammatory mediators such as cytokines, including interleukin-6 (IL-6), with the consequent generation of reactive oxygen species (ROS), leading to various gastrointestinal disorders, such as gastric ulcers [5,6].

Current pharmacological therapy used for treating gastric ulcer-related conditions has limited efficacy and is frequently associated with severe side effects; some drugs for this purpose belong to the proton pump inhibitors, histamine (H2) receptor antagonists, and anticholinergics [7,8,9,10,11]. In the search for alternatives, pre-clinical research of medicinal species and their products is carried out. In this sense, Mexico is considered a country with a broad extension of ecosystems, where floristic diversity is immense, which adds to its vast culture in its medicinal use.

For example, the flowers of *Malvaviscus arboreus* are widely used in Mexico by ethnic groups in the preparation of salads, herbal tea, and herbal dyes; furthermore, it is also used for the treatment of cystitis, diarrhea, sore throat, cold, bronchitis, thrush, tonsillitis, fever, and mainly in gastric disorders. Known in Mexico as “Molinillo,” “Monancillo,” “Manzanita,” “Sibil,” “Mazapan,” and “Malvavisco” [12], it is an erect and perennial shrub that extends throughout the USA, Mexico, Central, and South America.

Pharmacologic reports show that *M. arboreus* exhibited acts in an insect-repellent, molluscicide, anti-thrombotic, antitussive, antibacterial, antioxidant, antifungal, and hepatoprotective capacity [13,14,15,16,17,18,19,20]. This plant has different kinds of chemical constituents such as fatty acids (octadecadienoic acid, nonadecadienoic acid derivatives), phenol acids (gallic, protocatechuic, and p-hydroxybenzoic), hydroxycinnamic acids (chlorogenic, p-coumaric, ferulic and synaptic), and flavonoids (cyanidin, kaempferol, and apigenin) [16]. Recently, two glycosylated flavonoids, present in a fraction with anti-ulcerogenic activity, identified as kaempferol 3-*O*-sophoroside and kaempferol 3-*O*-sambubioside, were isolated [21]. These two compounds were obtained as a mixture, called MaSS, and the ethanol-induced gastric injury assay was used to further their study of their gastroprotective effect. This model of gastric lesions, induced by ethanol, is widely used in rodents; it is known that the excessive intake of ethanol causes gastric damage; the administration of this substance acutely activates several mechanisms, such as the exposure of gastric tissue to the actions of hydrochloric acid and pepsin, reduces blood flow, and causes microvascular injuries by increasing the production of reactive oxygen species (ROS) and proinflammatory cytokines, thereby reducing levels of natural antioxidants [22,23,24].

Although the pro-oxidant environment is generated continuously, under normal physiological conditions, the organism can counteract it through a well-known antioxidant defense system composed of enzymatic and non-enzymatic mechanisms. The first cellular system comprises the enzymes superoxide dismutase (SOD), glutathione peroxidase (GP), glutathione reductase (GR), catalase (CAT), and superoxide reductases (SR) [6]. Different substances have been used as pharmacological standards in the assay of gastric lesions caused by ethanol; among them, L-Arginine (L-Arg), an amino acid that has been used experimentally as a precursor of nitric oxide, can enhance the antioxidant capacity of cells [25], and has demonstrated its effectiveness in protecting the stomach against lesions induced with ethanol; its effect depends on the dose used, since doses less than 300 mg/kg have no effect [21,26].

According to the data mentioned above, this work aimed to extend the knowledge about the preventive capacity of MaSS from *Malvaviscus arboreus* for diminished gastric damage induced by the administration of ethanol through acts on biochemical and histological parameters such as relative weight of the stomach, gastric lesions (%), concentration of local cytokines IL-6 and IL -10, gastric CAT activity, and the hematoxylin-eosin staining technique (evaluation of the edema in different substrate stomachs).

## 2. Results

### 2.1. Chemical Analysis

A chromatographic analysis indicated that the treatment of MaSS was constituted by two major compounds eluting at 8.9 min and 9.2 min, with a classical wavelength of flavonols (265,350 nm, Figure 1). The comparison of treatment MaSS with previously isolated flavonoids from flowers of *M. arboreus* allowed us to identify them as kaempferol-3-*O*-sophoroside and kaempferol-3-*O*-sambubioside at 8.95 min and 9.21 min, respectively [10,19]. The proportion of these glycosylated flavonoids in the MaSS mixture was 76% kaempferol-3-*O*-sophoroside and 24% kaempferol-3-*O*-sambubioside.

### 2.2. Effect of MaSS on Stomach Weight

Figure 2 shows the percentage of stomach weight that corresponds to the total animal weight, so that animals with absolute ethanol, without treatment (Veh), showed a higher percentage of weight with 0.763 ± 0.136 than the healthy group, with a value of 0.426 ± 0.039 (^#^
*p* < 0.05). The animals that received the positive control drug, L-Arg (0.413 ± 0.009), or MaSS at 30 mg/kg (0.524 ± 0.040), 90 mg/kg (0.499 ± 0.049), 120 mg/kg (0.473 ± 0.052), or 180 mg/kg (0.540 ± 0.051), had a significant decrease in this variable compared to the Veh group (* *p* < 0.05).

### 2.3. Effect of MaSS on the Ulcerated Stomach

Figure 3 shows that the Veh group had a 46.65 ± 2.65% of the ulcerated area, which is significantly different from the group of healthy animals (0.0%) since they do not have gastric damage (^#^
*p* < 0.05). All of the experimental treatments, including L-Arg at 300 mg/kg with an ulcerated area of 1.357 ± 0.85%, and MaSS at 30 mg/kg with 3.77 ± 0.92%, 90 mg/kg with 4.01 ± 1.1%, 120 mg/kg with 0.52 ± 0.097% and, finally 180 mg/kg with 7.82 ± 3.0%, induced all, a significant inhibition of the percentage of gastric lesion when compared to Veh (* *p* < 0.05).

Figure 4 presents photographs of stomachs from each experimental group. The first one is representative of healthy rats (Figure 4A), so they do not show gastric mucosal damage. Figure 4B corresponds to the Veh treatment and shows that animals administered with Veh and ethanol had the highest percentage of gastric lesions. 

The administered treatments, L-Arg and MaSS, reduced the damage caused by ethanol (Figure 4).

### 2.4. Histological Analysis 

Figure 5 shows histological sections obtained with a Model LSM 800 confocal laser scanning microscope, Carl Zeiss (Munich, Germany) and taken 5x in TIFF format (2048 × 2048 pixels). Samples were mounted on glass slides and were observed in lambda mode in which a sequence of images was collected at laser wavelengths of 405 nm, 488 nm, 561 nm, and 640 nm (4% capacity). The ZEN software version 2.6, Zeiss Blue edition was used. Samples were taken using a coupled HD camera (AxioCam, Carl Zeiss, Model 305, color, Oberkochen, Germany).

The stomach of healthy rats shows an unaltered architecture, with firmness and regularity in the tissue, without infiltrating leukocytes or epithelial damage (Figure 5A). The group with gastric ulcers (Veh, Figure 5B) has a tissue where severe bleeding is observed, such as the infiltration of leukocyte cells, mainly lymphocytes, plasma cells, and eosinophils with pyknosis nuclei, epithelial and glandular tissue destruction, in addition to cellular irregularity. Mucin erosion was observed in the mucosa, along with prominent dilatation of the muscularis mucosae and submucosa. Degeneration of the circular and longitudinal layers of the tunica muscularis provoked anincrement of the sinusoidal spaces.

Treatment with L-Arg decreases necrosis and damaged tissue, with moderate mucosal dilation (Figure 5C); the mixture of kaempferol 3-*O*-sophoroside and kaempferol 3-*O*-sambubioside (MaSS) at different doses (Figure 5D–G) decreases the distance of the sinusoidal spaces of the mucosa and the necrotic damage in this tissue is lower than that of the Veh group (Figure 4B); no epithelial or glandular destruction is observed, with an appreciable increase in mucin, slight dilatation of the mucosa and submucosa, less density of infiltrated cells and the tunica muscularis circular layer shows only slight degeneration.

Figure 6A shows that the gastric mucosa thickness of healthy animals was 20.09 ±1.44 µm; the group of rats exposed to ethanol and only treated with vehicle exhibited edema of 45.397 ± 1.07 µm; both groups were statistically different (^#^
*p* < 0.05). The administration of L-Arg diminished the thick mucosa to the value of 21.31 ± 1.11 µm. Different doses of MaSS, 30 mg/kg (20.9 ± 0.29 µm), 90 mg/kg (24.3 ± 0.09 µm), 120 mg/kg (24.3 ± 1.03 µm), and 180 mg/kg (13.81 ± 1.06 µm) significantly diminished ethanol-induced edema; all of these groups were significantly different from Veh (* *p* < 0.05).

The thickness of the muscularis mucosae of those healthy animals was 3.7 ±0.09 µm), which is significantly lower (^#^
*p* < 0.05) than that of the group of animals with ethanol-induced damage (Veh, 8.232 ± 0.07 µm). Although all treatments, including L-Arg with a value of 6.18 ± 0.07 µm, MaSS at 30 mg/kg (6.18 ± 0.012), MaSS at 120 mg/kg (6.89 ± 0.28 µm), MaSS at 180 mg/kg (6.12 ± 0.32 µm) showed decreased edema compared to the Veh group (* *p* < 0.05), the dose of 90 mg/kg induced the most significant activity with a thickness of 2.95 ± 0.094 µm (Figure 6B).

Ethanol induces a similar effect on the stomach submucosa; thus, the healthy group has a thinner layer (with a value of 16.62 ± 0.32 µm) than the damaged stomach that had edema of 31.3 ± 1.4 µm, showing a significant difference between both groups (^#^
*p* < 0.05). The effect produced by the ethanol on the gastric tissue was ameliorated with the administration of L-Arg (15.4 ± 0.72 µm), and with MaSS at 30 mg/kg had submucosal edema of 21.61 ± 0.045, at 90 mg/kg was of 13.6 ± 0.86, 120 mg/kg caused a level of 14.761 ± 0.8, and 180 mg/kg of 16.9 ± 2.01 µm, all these were significantly different from Veh (Figure 6C, * *p* < 0.05).

### 2.5. Cell Count of Each of the Stomach Strata

Cell counts were carried out in different strata of the stomach. Ethanol (Veh) causes a significant decrease in the number of cells in the three layers, mucosa, muscularis mucosae, and submucosa, compared to the stomach of healthy rats (^#^
*p* < 0.05, Table 1). On the contrary, with the administration of L-Arg, the number of cells in the three strata was significantly higher than in the Veh group (* *p* < 0.05). Furthermore, all doses of MaSS significantly increased the number of cells in the submucosa; the doses of 90 mg/kg produced this action in the mucosa and muscularis mucosae, and 180 mg/kg provoked an increase of mucosal cells (* *p* < 0.05).

### 2.6. Effect of MaSS on IL-6, IL-10, and Catalase

Figure 7A shows that ethanol caused a significant increase (^#^*p* < 0.05) in the gastric concentration of IL-6 with a value of 859.2 ± 37.5 pg/g protein, compared to the healthy group whose value was 86.4 ± 56 pg/g protein. The drug L-Arg counteracted this effect by reducing the local concentration of that protein with 195.04 ± 140.6 pg/g protein. The administration of MaSS produced a diminution of the IL-6 gastric, for 30 mg/kg a concentration of 631 ± 46.09 pg/g protein was observed, while with 90 mg/kg it was of 285.5 ± 157 pg/mg protein, for 120 mg/kg it was 668.76 ± 47.4 pg/g protein, and for 180 mg/kg it was 308.4 ± 112.9 pg/g protein. The behavior of the flavonoid mixture varies between the groups, although all were significantly different from the Veh group (* *p* < 0.05). 

In Figure 7B, the concentration of IL-10 in the stomach of rats that received ethanol (Veh group) was 39.58 ± 11.7 pg/g protein, which was significantly lower compared to the healthy group with 496.7 ± 82.4 pg/g protein (^#^
*p* < 0.05). The administration of the different treatments in L-Arg (401.3 ± 46.9 pg/g protein) and MaSS at 30 mg/kg (351.3 ± 65.5 pg/g protein), at 90 mg/kg (518.2 ± 77.6 pg/g protein), at 120 mg/kg (1750 ± 481 pg/g protein), and at 180 mg/kg (1669.04 ± 249 pg/g protein), all of these data were significantly elevated in comparison with the Veh group (* *p* < 0.05).

The ethanol administration provoked a CAT activity of 0.0077 ±0.001 U/mL, data that was significantly lower (Figure 7C, ^#^
*p* < 0.05) than the animals of the healthy group with values of 0.0217 ± 0.0012 U/mL. The control drug, L-Arg, counteracted the effect of ethanol with a significant difference between this group and the Veh group (* *p* < 0.05) because, in this group, the CAT activity was 0.0099 ± 0.0001 U/mL. The treatment of the animals with MaSS at 30 mg/kg presented values of 0.0183 ± 0.0017 U/mL, which was reducing at 90 mg/kg 0.014 ±0.0042 U/mL, at 120 mg/kg 0.0140 ±0.0020 U/mL, and at 180 mg/kg of 0.0119 ± 0.0027 U/mL; all groups with MaSS have a statistical difference with the damage and healthy groups (* *p* < 0.05).

### 2.7. Damage Scores in Gastric Lesions Induced with Ethanol

Table 2 shows the scores of each variable recorded; the sum of them in the Veh group was 30 (Table 2), while in the basal group it was 0. The protective effect of L-Arg and MaSS at different doses was observed due to the scores less than those of the damage group.

Each symbol (O Figure 8) represents ten variables of damage evaluated and how many of these add to the score indicated on the Y axis. As shown in the methodology, the maximum damage score in each parameter is 3, and without damage it 0. Thus, for the group of healthy rats, the damage score for each variable was 0, while for Veh, it was 3 in all parameters; both groups were statistically different from each other (*^#^
*p* < 0.05, Figure 8). For the L-Arg group, it is observed that it has a score of 0 for most of the variables and only one variable with a score of 2, and shows a statistical difference with Veh (* *p* < 0.05).

The response of the rats to the administration of different doses of MaSS is varied. For example, MaSS at 90 mg/kg presented two variables with a score of 1, and the others were 0. For the 120 mg/kg, three variables had a score of 2, one variable had a score of 1, and the rest had a score of 0; for 180 mg/kg there was one variable of 2, three of 1, and the others were at 0; these groups were different from Veh (* *p* < 0.05). Finally, the lower dose of 30 mg/kg has three variables with a score of 2, five with a score of 1, and only two with a score of 0; the statistical data indicated that between this dose and the Veh, there was no statistical difference (*p* > 0.05).

## 3. Discussion

Gastric lesions are part of the pathophysiology of GU, which represents a widespread disorder that is multifactorial and that has a high burden on health care systems around the world. [27,28,29]. There are different models of induction of gastric lesions, including those produced by the acute administration of ethanol. In this assay, oxidative stress is generated by the release of free radicals due to the deregulation of the H^+^/K^+^ ATPase pump, which leads to mucosal damage with hemorrhagic erosions and ulcers over time; one to two h after alcohol administration, the percentage of damage is between 10 and 40% [8].

The mixture (MaSS) of kaempferol 3-*O*-sambubioside and kaempferol 3-*O*-sophoroside [21] was evaluated at different doses, with the objective of determining if this natural product could prevent the formation of a gastric lesion in rats to which ethanol was administered. Results indicated that this could act on different levels of the lesions. For example, the stomach size (relative to body weight) was measured as a damage variable, as a weight percentage. It was observed that, in the ulcerated rats, it was 0.76% significantly higher than the group without ulcers (healthy, 0.42%); the administration of MaSS diminished that value at different doses compared with the Veh group.

In the literature, there is no data about this variable (percentage of relative weight) in the assay used here; however, we consider it significant since the inflammation and hemorrhagic edema associated with ulcer lesions are causing an increase in the weight of the organ, which is a gross measurement of inflammation. This variable is consistent with the effect of gastro-protective agents because the stomach of animals who received different doses of MaSS reduced the percentage of gastric lesions with values of only 0.5 to 7% compared to the Veh group, which presented 46% of ulcerate area. This data is consistent with the literature, where it is mentioned that ethanol causes damage in 10 to 40% of the stomach [8,29,30,31,32]. The mixture MaSS is flavonoid derivatives of kaempferol, a compound that, administered at doses of 0.1, 0.3, and 1 mg/kg, was able to reduce the percentage of the ulcerated area of the stomachs of rats that received ethanol/HCl [3]. In another study, kaempferol at doses of 50 and 100 mg/kg had a gastro-protective effect; however, activity was lost when the dose was increased to 250 mg/kg [4]. Even when MaSS from *M. arboreus* protects the stomach from damage, a similar behavior was observed since high doses (180 mg/kg) induced less effect than low doses in several of the parameters analyzed.

Ethanol is an agent of damage to the gastric mucosa, as pointed out throughout this work. Excessive intake of this substance is known to result in a "surge" of activated neutrophils that infiltrate the site of injury, leading to damage by the increased production of pro-oxidant agents, free radicals, and pro-inflammatory molecules such as cytokines [33]. A consequence of gastric damage is the recruitment of leukocytes that stimulate the inflammatory response. Different studies have shown that they release pro-inflammatory cytokines such as tumor necrosis factor-alpha (TNF-α), IL-1β, and IL-6, which importantly regulate gastric lesion production [5], and the levels of these markers are significantly elevated in that condition [34]. In the present work, the concentrations of IL-6 and IL-10 were quantified. It was observed that the animals of the Veh group presented a high stomach concentration of IL-6, with a significant decrease in the anti-inflammatory molecule, as indicated in other reports [5,35]. 

All doses of MaSS could counteract this effect of ethanol, so it can be proposed that MaSS also acts as a modulator of the immune response by modifying the inflammatory process associated with IL-6. This molecule is multifunctional and is considered a regulator of acute inflammation, as is the case of ethanol-induced gastric lesions; and it also regulates chronic inflammation. It can stimulate neutrophils, macrophages, and lymphocytes at the site of inflammation, thereby releasing harmful products such as oxygen free radicals, lysosomal enzymes, and cytokines, which are responsible for the damage to the gastric mucosa [36].

With these data, it is possible to infer that part of the mode of action of MaSS as a preventive agent against gastric lesions is due to its actions on the local concentration of cytokines IL-6 and IL-10. The chemical precursor of MaSS is kaempferol, which is capable of reducing IL-6 levels in people with peptic ulcer disease, which is a risk factor for stomach cancer [36].

Furthermore, ethanol causes an increase in ROS production in the gastric mucosa, causing the mucus layer to be eliminated and cell death to take place. However, the gastric mucosa maintains its function and structure due to the balance between aggressive and protective factors, (SOD, CAT-, glutathione reductase -GSH-, among others). The over-production of ROS in the ulcerated stomach is associated with a decrease in the activity of these enzymes. CAT mainly fulfills the function of converting hydrogen peroxide (generated by the action of SOD on superoxide radicals) into water and oxygen. In the present work, the MaSS mixture protect of oxidative stress by modulating the CAT response to damage.

This microscopic study of the effects of MaSS on ethanol-induced gastric lesions is essential because it summarizes the overall beneficial activity that the flavonoid mixture is causing so that the damage to gastric histoarchitecture is less than the Veh group after administration. Furthermore, greater integrity of the epithelium and mucosa was observed, there were fewer areas of edema and infiltration of inflammatory cells such as neutrophils and minimal bleeding. In addition, this study allowed the analysis of the level of microscopic edema in different gastric layers, showing that inflammation decreases in the mucosa, submucosa, and muscularis mucosae due to treatment with this natural product, in congruence with previous results.

Regarding L-Arg, this amino acid has a gastroprotective effect; it has been established that it depends on the dose used. For example, 100 mg/kg of L-Arg does not prevent ulcerative lesions, but 300 mg/kg of this amino acid can antagonize ethanol’s effect. Therefore, this was the reason for the use of that dose. Furthermore, it has been mentioned that the gastro-protective effect of L-Arg is due to its ability to increase the concentration of nitric oxide (NO), which is a potent vasodilator, and that this molecule plays a fundamental role in gastric hemostasis by modulating the basal tone of the vasculature, increasing mucosal blood flow, regulating mucus and bicarbonate secretion, inhibiting gastric secretion, and protecting the mucosa against damage induced by a wide variety of corrosive substances [26]. However, other authors propose that the positive actions of L-Arg when orally administered are attributed to its cytoprotective effect by acting on prostaglandins rather than being an NO precursor and thus activating the functions of this gas in the gastric vasculature. The effect of L-Arg on these inflammatory mediators appears to be blocked by co-treatment with indomethacin, a prostaglandin inhibitor [37]. The administration of 300 mg/kg of L-Arg decreases the concentration of IL-6 and increases IL-10, from which it is deduced that this amino acid exerts a modulating activity of the response associated with inflammation in the stomach of rats that received ethanol as an agent of gastric damage.

In the current literature, there are no data on this effect of L-Arg in models of gastric ulcers; however, this result is consistent with that cited in other works, on the actions of this amino acid on cytokines, in different models of diseases and clinical assays. For example, in a mercury-induced toxicity model in Balb-C mice, there was an increase in the concentration of IL-6 in the spleen; supplementation with L-Arg decreased these values [38].

Finally, the results presented are relevant since they indicate that MaSS obtained from *M. arboreous* prevents damage to the gastric mucosa against the harmful effects of ethanol. This treatment can act as a modulator of the local inflammatory response, controlling the elevation in the gastric concentration of IL-6 and a decrease in IL-10. It acts through a preventive effect of the oxidative response caused by this alcohol, which reduces the damage to the stomach mucosa. These data serve as necessary background to promote further pharmacological studies using an experimental design in which MaSS is administered after the induction of gastric damage. In addition, other models of gastric ulcers can include, for example, gastric disease induced with *Helicobacter pylori*, in which this mixture of flavonoids could probably exert antiulcerogenic actions. Other in vivo and in vitro models could then permit us to evaluate its mechanism of action using different methods, such as PCR and molecular docking. It is important to carry out toxicity experiments with MaSS and increase the pharmacological knowledge of this treatment, in order to have a gastroprotective phytomedicine in the future.

## 4. Materials and Methods

### 4.1. Plant Material and Extract Preparation

Flowers of *Malvaviscus arboreus* Cav. (3 kg), were collected from a controlled culture at the Centro de Investigación Biomédica del Sur-IMSS, Morelos, Mexico. The voucher specimen of this material (No. 34413) was deposited in the herbarium of the Autonomous University of the State of Morelos, Mexico, identified by the taxonomist Gabriel Flores Franco.

The flowers were dried under dark conditions at room temperature for two weeks. This plant material was reduced to an average size of 4–6 mm in diameter in an electric mill (Pulvex, CDMX, City of Mexico, Mexico) [21], and a maceration process extracted 450 g with acetone (each 100 g of plant for 2 L) over 24 h. The acetonic extract was obtained by evaporation until reaching dryness by low-pressure distillation using a rotary evaporator (Heidolph G3, Schwabach, Germany), resulting in 13.5 g of solid extract which was adsorbed in silica gel and placed on a column of silica gel for gravity (20 g, 60F254, Merck, Darmstadt, Germany). A gradient of dichloromethane/methanol was used as the mobile phase, collecting 50 fractions of 30 mL each.

These fractions were concentrated in a rotary evaporator under reduced pressure, grouped, and identified with the help of TLC and HPLC. The one containing most flavonoids and sugars (3 g) from the grouped fractions was taken to another column but used reverse silica gel (10 g, RP-18F254, Merck, Darmstadt, Germany). A gradient of water/acetonitrile/methanol was used as the mobile phase, collecting 30 fractions of 10 mL each. These fractions followed the same process with the difference that the one containing a majority of kaempferol glycosides (3 g) was taken to another reverse column with the same gradient. They collected until obtaining a mixture of two compounds: kaempferol-3-*O*-sambubioside and kaempferol-3-*O*-sophoroside (MaSS).

### 4.2. High-Performance Liquid Chromatography (HPLC) Analysis of the MaSS Mixture

The chromatographic analysis was performed using an HPLC system equipped with a Waters 2695 separation module and a Waters 2996 Photodiode detector (Waters, Milford, MA, USA). Samples of 10 µL (1mg/mL) of MaSS were separated in a reverse phase Supelcosil LC-F column (250 mm × 4 mm, 5 µm particle size) (Merck, Darmstadt, Germany) connected to a guard column. The mobile phase consisted of a gradient system that was comprised of 0.5 % trifluoroacetic acid (solvent A) and acetonitrile (solvent B) as follows: 0–1 min, 0% B; 2–3 min, 5% B, 4–20 min, 30% B; 21–23 min, 50% B 14–15 min; 24–25 min, 80% B; 26–27 100% B; 28–30 min, 0% B, with a flow rate of 0.9 mL min^−1^. Absorbance was measured at 350 nm to identify these kaempferol disaccharides. Both flavonoids were identified by comparison with data for previously isolated compounds [21].

### 4.3. Gastric Lesions-Induced by Ethanol

#### Animals

Sprague Dawley Male rats from Centro Médico Nacional Siglo XXI (City of Mexico, Mexico), were used. The groups were formed with eight individuals each (weight range of 350–500 g). The animals were kept in a light-dark cycle of 12 h by 12 h. The temperature was 25 ± 2 °C and there was a constant flow of air. The feed was free of additives, hormones, antibiotics, drugs, pesticides, and pollutants, the rats had free access to water. The experimental animals were handled according to the official Mexican standard: NOM-062-ZOO-1999. The rats were kept in acrylic boxes 50 cm long, 40 cm wide, and 20 cm high. This experimental protocol was approved by the ethics and research committee at the Instituto Mexicano del Seguro Social, with the registration number R-2018-1702-015.

### 4.4. Treatments

The experiments were done according to the Morimoto method [38]. Rats were randomly divided into groups which were subject to different treatments that were administered orally (per os, p.o.).

Group 1. Healthy rats without gastric lesions treated with Tween 20 only, (1 %) by p.o.

Group 2. Vehicle animals with gastric lesions induced with ethanol and treated with Tween 20 (1%) by p.o.

Group 3. Amino acid L-arginine rats with gastric lesions induced with ethanol and treated with L-Arg, 300 mg/kg, p.o. as the control group; this drug reduces oxygen free radicals and inhibits the production of COX-1 and COX-2 enzymes in charge of vasodilating stomach tissues [39].

Groups 4–8. MaSS from *M. arboreus* at different doses: 30, 90, 120, and 180 mg/kg, by p.o., based on a previous report [21], in which 60 mg/kg was used, and therefore a curve was proposed that included multiples of the first dose.

### 4.5. Gastric Ulcers Murine-Model

After 48 h of fasting, the animals received their respective treatment, and 1 h after this, they were administered (by p.o.) with 1 mL of absolute ethanol per 200 g weight to induce gastric lesions. After this, the rats were euthanized by an overdose of intracardial urethane [40,41], and the stomachs were removed and opened by the mean curvature. Stomachs were photographed with a digital camera (Canon EOS 70D (W), Tokyo, Japan) in a dark box with a zoom of 55×. The area of mucosal injury was measured using ImageJ 1.44p software (National Institutes of Health, Bethesda, MD, USA) [42].

Of the eight stomachs of each group, five were stored at −70 °C and were homogenized. This was used to quantify the activity of the enzyme catalase and the cytokines IL-6 and IL-10; the remaining three were immersed in 10% formaldehyde/buffer (volume/volume) for histological analysis.

#### 4.5.1. Determination of the Ulcerated Area

Captured images of the stomachs (Canon EOS 70D (W), Tokyo, Japan) for each treatment were used to determine the ulcerated area. For this purpose, the ImageJ V1.44 program used a color threshold plugin to contrast and segment the ulcerated area.

The ulcerated area percentage was calculated using the same plugin to contrast the staining color, and then it was calculated with the following equation [43,44]:Ulcerated area **=** Number of pixels of ulcerated area × 100/number of pixels of total stomach (1)

#### 4.5.2. Determination of the Microscopic Edema

I Images of each treatment were obtained in the confocal scanning microscope at 5X objective with a numerical aperture of 0.5, it was used a bright field as a contrast technique, and the edema area of each of the stomach strata (mucous membrane, muscularis mucosae, and submucosa) was quantified using Image J software. The micron quantification was done by limiting each image’s specific area. All of the strata areas were standardized (0.75 × 1.16 mm), and a limited image was marked manually with free reference lines to delimit the stratum and differentiate it; the value obtained in microns is defined as edema.

#### 4.5.3. Determination of the Cell Count

Using the stomach strata images of each of the treatments obtained in the confocal scanning microscope at 5× with a numerical aperture of 0.5 bright fields and the Image J program, we quantified the number of cells present in a section of 200 mm^2^ for each of the stomach layers (mucosa, muscularis mucosa, submucosa). The quantification was done by limiting them to a specific area in each image so that all of them were standardized in the same section. This limited image was biased using the threshold plugins of the Image J software only to have the cells density that was counted present in the image.

#### 4.5.4. Quantification of CytokinesIL-6, and IL-10 by ELISA, and Catalase Activity

The rats’ stomachs were dissected and frozen, further disintegrated into a phosphate buffer (pH 7.0) with protease inhibitor (PMFS), centrifuged for 5 min at 10,000 rpm, and the supernatant was collected and stored at -70 °C. In these samples, the cytokines IL-6, and IL-10 and the catalase activity were measured. The measurement technique was performed using a kit (OptEIATM ELISA sets; BD Biosciences, Franklin Lakes, NJ, USA) according to the manufacturer’s instructions. Briefly, to 96-well plates, we added 100 µL/well of the antibody uptake; the plates were incubated for 12 h at 4 °C.

Once this time had elapsed, the dish was washed with PBS (Phosphate Buffer Saline) solution (0.05% of Tween-20, 300 μL/well × three times). Next, we added 100 μL of PBS with Fetal Bovine Serum (FBS) at 10%, pH 7.0 for 1 h at room temperature.

The contents were discarded, and the plate was washed with PBS buffer (0.05% of Tween-20, 300 μL/well × three times). To the corresponding wells we added 100 μL of the standard, the blank (PBS with FBS) and the test samples. The plate was incubated for 2 h at room temperature. The contents were discarded, and the plate was washed with PBS buffer (0.05% of Tween-20, 300 µL/well × five times). The detection antibody and a Streptavidin-Horse Radish Peroxidase (HRP) enzyme solution was added. These plates were incubated for 1 h and washed with 300 μL/well × seven times with a PBS solution (combined with 0.05% of Tween-20).

To each well, 100 µL of O-Phenylenediamine (OPD) previously prepared substrate was added (one tablet of OPD and one of urea dissolved in 20 mL of distilled water).

Afterward, this was incubated for 30 min at room temperature under total darkness, and we added a stop solution (H2SO4, 2N). Finally, the plates were read in a Stat Fax 2100 spectrophotometer (Awareness Technologies, Bellport, NY, USA) at a 450-nm wavelength at 37 °C.

Catalase activity was determined using the Catalase Assay Kit (Sigma-Aldrich, St. Louis, MI, USA), with 100 µL of the enzyme extract (homogenate) in 50 mM phosphate buffer (pH 7.0) monitoring the decrease in absorbance at 240 nm for 30 s after the addition of 10 mM hydrogen peroxide (H2O2). One unit of enzymatic activity is the amount of enzyme present capable of decomposing 1 mM H2O2/min at 25 ° C; in both cases, the specifications described by the suppliers were followed.

#### 4.5.5. Histology of Stomach with Gastric Ulcers

The stomachs were stored for one week in 10% formalin until used. For histological analysis, the stomachs were cut into six parts, placed in cassettes, embedded in paraffin, cooled to generate cubes that were cut with a microtome, and then recovered on a slide, which was stained with hematoxylin-eosin (HE). The stomach sections were observed at 5× objective in a confocal scanning microscope with an operational mode of bright field. 

#### 4.5.6. Score of Damage

The effect of MaSS on the global damage caused by absolute ethanol was categorized using a level based on the arbitrary assignment of a score in which 3 was the maximum value of damage observed for the different variables of the Veh group (model of damage). The parameters included were cell count of the mucosa, muscular of mucosae and submucosa; microscopic edema of the same strata; local levels of IL-6, IL-10; and also, relative stomach weight (%) and gastric ulcer (%). These scores were analyzed and summarized as the sum (Table 3), and the generated data was plotted.

### 4.6. Statistical Analysis

Statistical tests were applied according to each variable. For those that met a normal data distribution, analysis of variance (ANOVA) was used, accompanied by a Tukey’s post-test establishing the significance value of *p* < 0.05. Kruskal-Wallis analysis with Dunn’s multiple comparisons post-test was used for the relative weight (%) and gastric macroscopic lesions (%). Finally, the Friedman test was used with Dunn’s multiple comparisons test for damage scores. The symbols used for all cases were (^#^) to indicate statistical differences compared with the healthy group, and (*) was used for differences with the Veh group. The data is represented in box-and-whisker plots, scatter plots, and tables. This analysis was made using the statistical program Sigma Stat for Windows V.11.0.

## 5. Conclusions

Kaempferol-3-O-sambubioside and kaempferol-3-O-sophoroside can prevent gastric lesions, which are mediated by counteracting the variables associated with ethanol-induced gastric lesions, such as inflammation of the stomach (% weight), ulcers (%), the damage at the microscopic level, the dysregulation of cytokines, and catalase. According to the global effect of treatment with MaSS summarized as the damage score, the mixture MaSS was effective, primarily at doses of 90, 120, and 180 mg/kg.

## Figures and Tables

**Figure 1 plants-11-02951-f001:**
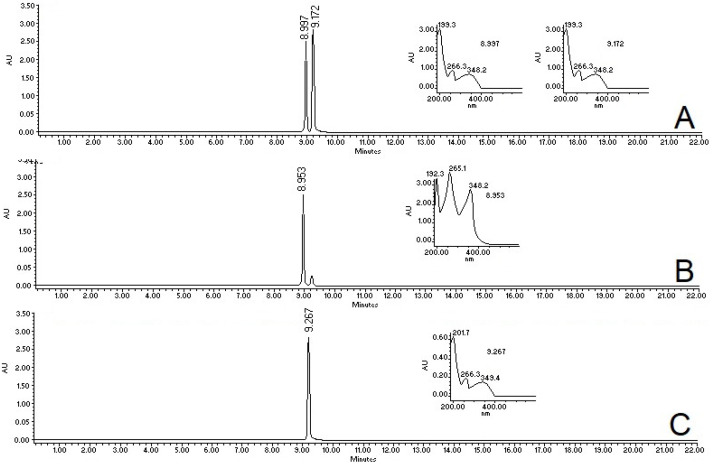
The fingerprint of treatment of MaSS (**A**), which was compared with previously isolated kaempferol-3-*O*-sophoroside (**B**) and kaempferol-3-*O*-sambubioside (**C**). All samples were recorded at l = 350 nm; AU = absorbance units.

**Figure 2 plants-11-02951-f002:**
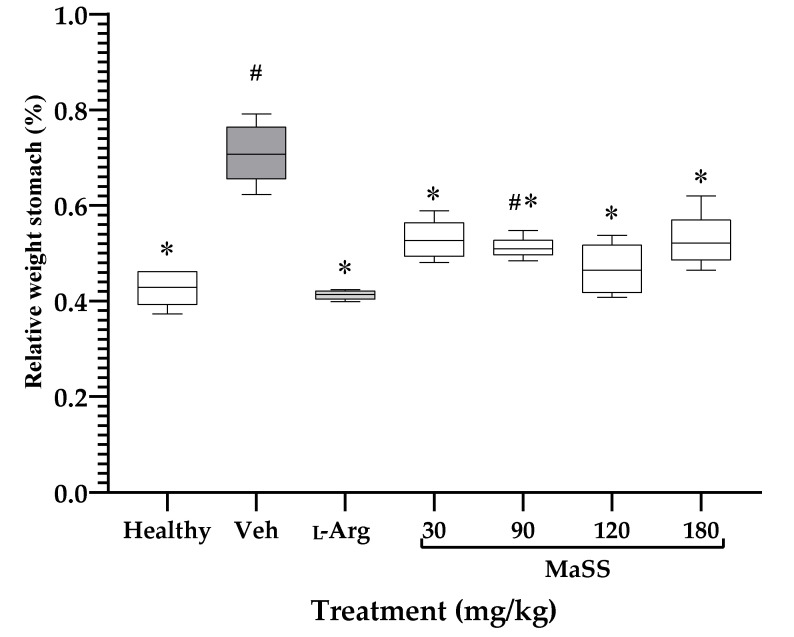
Effect of MaSS at different doses on relative stomach weight of rats with ethanol-induced gastric lesions. Healthy = rat without gastric lesions, Veh = rat with gastric ulcers, L-Arg = L-arginine (300 mg/kg), MaSS = mixture of kaempferol 3-*O*-sophoroside and kaempferol 3-*O*-sambubioside. Kruskal-Wallis with a post-test of Dunns’s multiple comparisons (n = 5, $\bar{x}$ ± ED, **^#^**
*p* < 0.05 when groups are compared with healthy group; * *p* < 0.05 compared with Veh group).

**Figure 3 plants-11-02951-f003:**
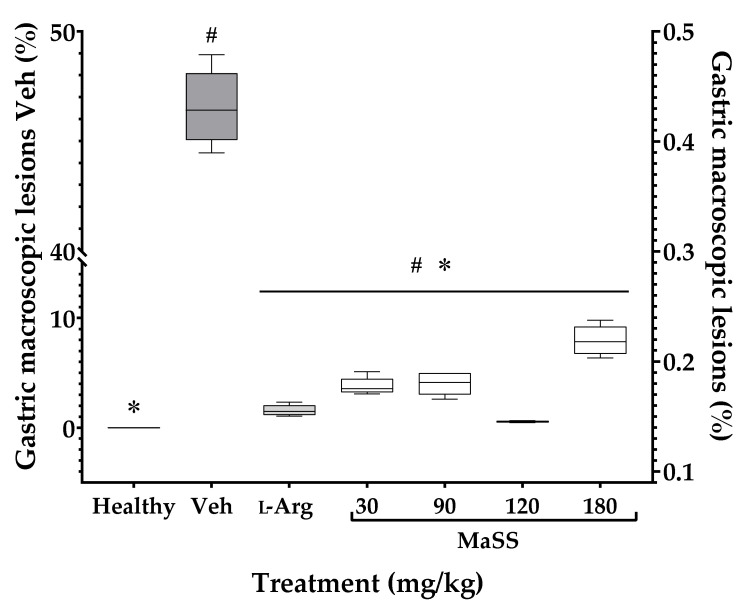
Effect of MaSS at different doses on the percentage of gastric ulcers induced with ethanol in rats. Healthy = rat without gastric lesions, Veh = rat with gastric lesions, L-Arg = L-arginine, MaSS = mixture of kaempferol 3-*O*-sophoroside and kaempferol 3-*O*-sambubioside. Kruskal-Wallis analysis with Dunn’s multiple comparisons post-test (n = 5, ± ED, ^#^
*p* < 0.05 when groups are compared with the healthy group; * *p* < 0.05 compared with Veh group).

**Figure 4 plants-11-02951-f004:**
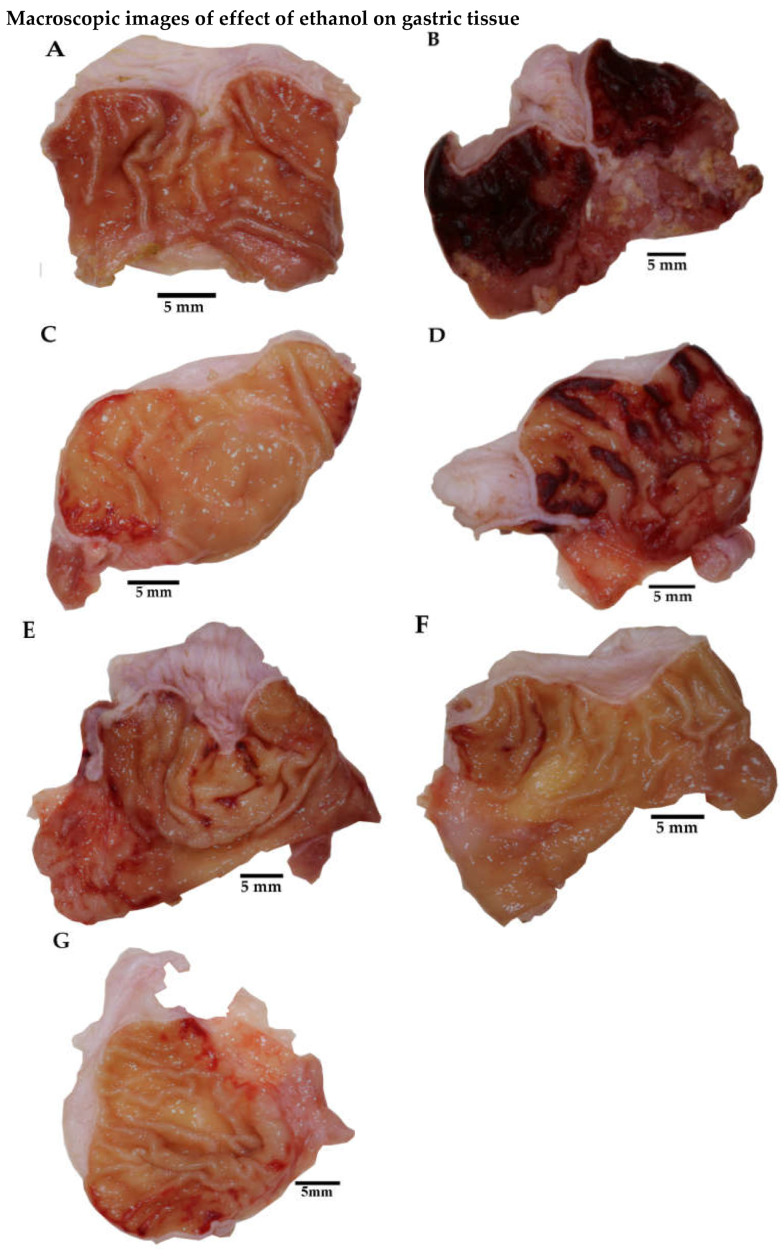
Photographic images show a representative stomach from each experimental group. (**A**) Healthy = rat without gastric lesions, (**B**) Veh = rat with gastric ulcers, (**C**) L-Arg = L-arginine, (**D**) MaSS = 30 mg/kg, (**E**) MaSS = 90 mg/kg, (**F**) MaSS = 120 mg/kg, (**G**) MaSS = 180 mg/kg is a mixture of kaempferol 3-*O*-sophoroside and kaempferol 3-*O*-sambubioside.

**Figure 5 plants-11-02951-f005:**
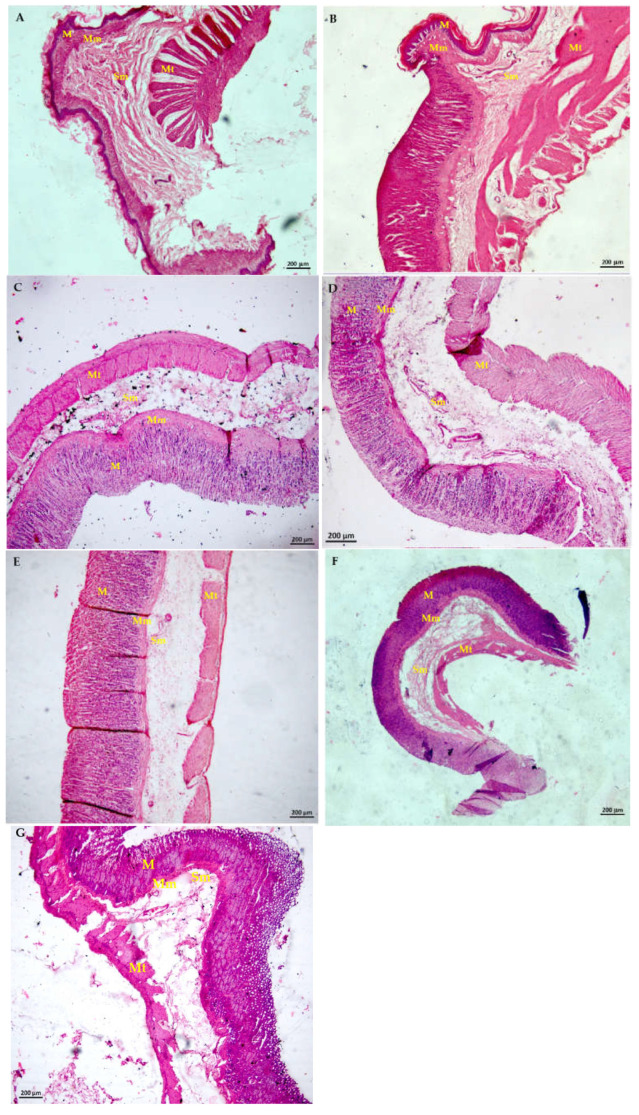
Effect of MaSS on the microscopic structure of the rat stomach with ethanol-induced gastric lesions. Photographs of histological sections observed at 5× objective in a confocal scanning microscope, with an operational mode of bright field (**A**): Healthy group; (**B**): Veh group with gastric ulcers with tween 20; (**C**): L-Arg = L-arginine; (**D**–**G**): MaSS to 30, 90, 120 and 180 mg/kg of a mixture of kaempferol 3-*O*-sophoroside and kaempferol 3-*O*-sambubioside. M = muscularis; Mm = muscularis mucosae; Sm = submucose; Mt = muscularis tunica.

**Figure 6 plants-11-02951-f006:**
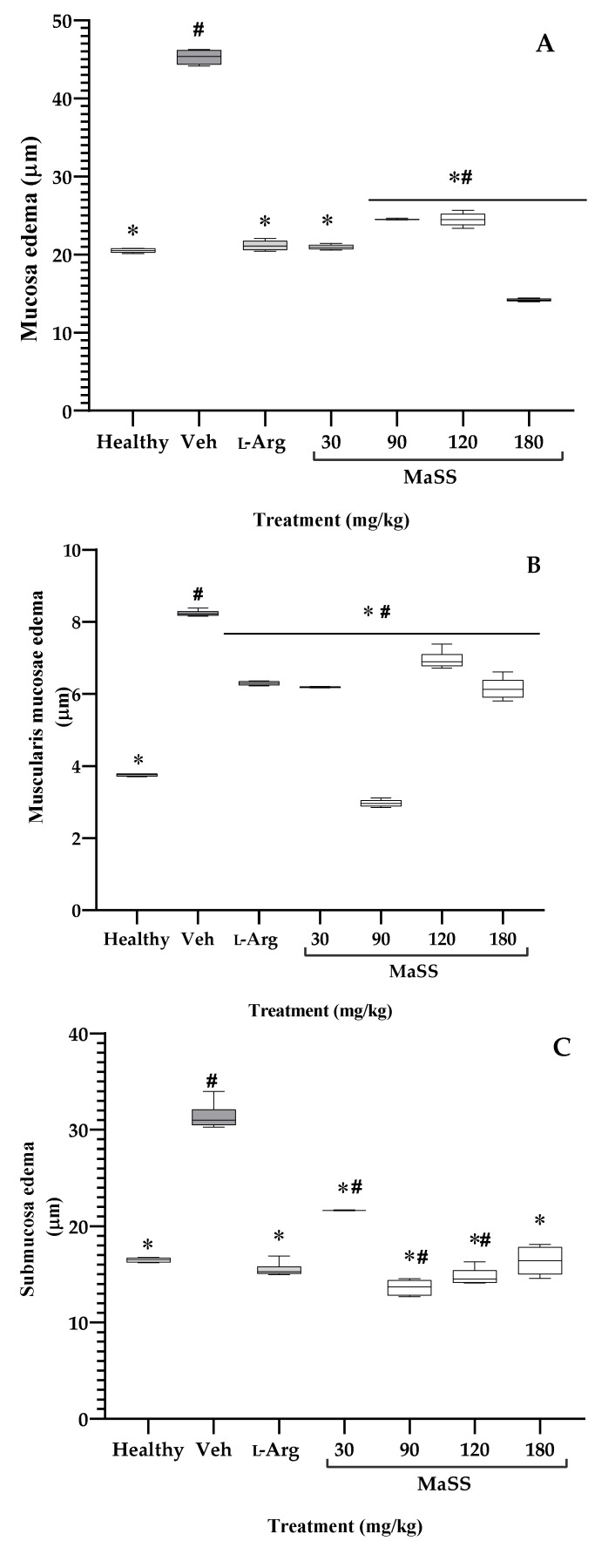
Effect of different doses of MaS, the mixture of kaempferol 3-*O*-sophoroside and kaempferol 3-*O*-sambubioside, obtained from *M. arboreus*, on microscopic edema of mucosa (**A**), muscularis mucosa (**B**), and submucosa (**C**) from the stomach of rats with gastric lesions-induced with ethanol. Healthy = rats without lesions; Veh = rats with gastric lesions. L-Arg = amino acid L-arginine. ANOVA post-Tukey’s test (n = 5, ± ED, ^#^
*p* < 0.05 when groups are compared with the healthy group; * *p* < 0.05 compared with Veh group).

**Figure 7 plants-11-02951-f007:**
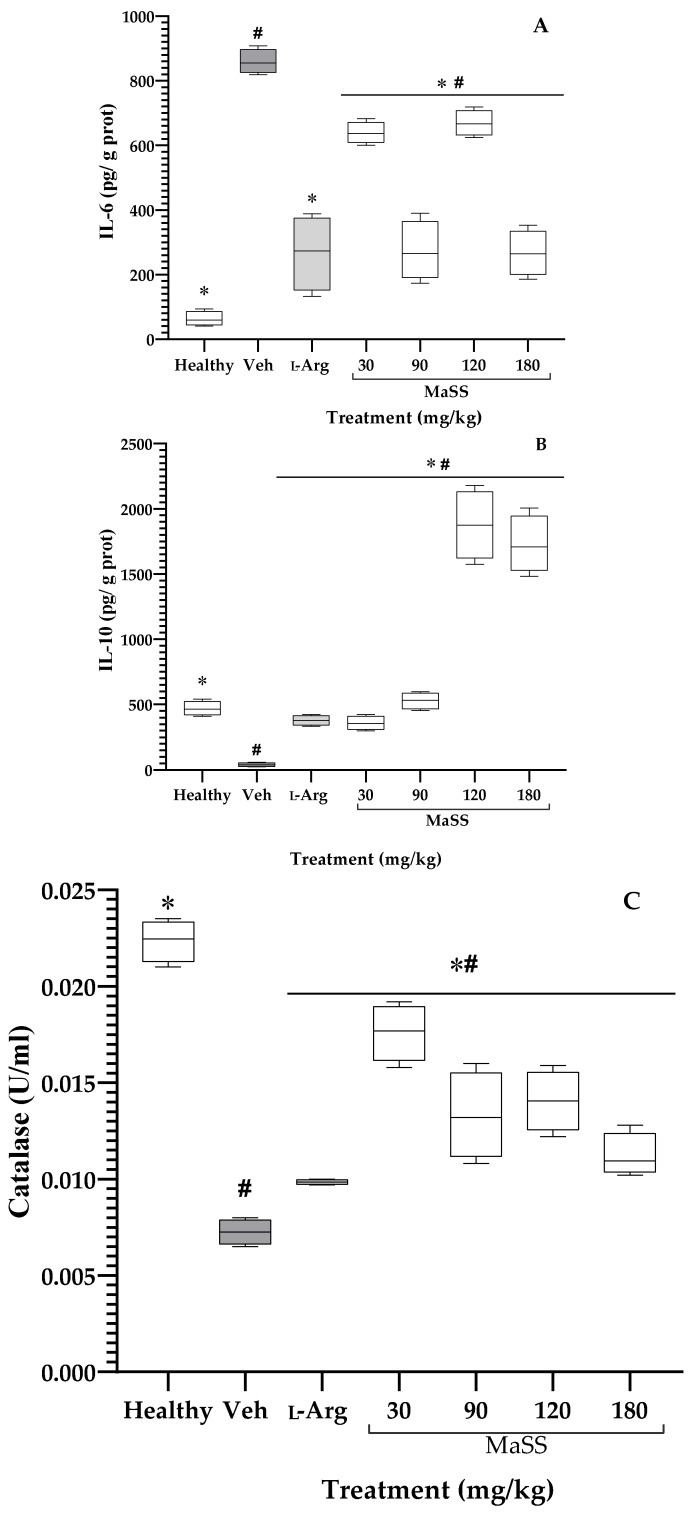
Effect of different doses of MaSS, mixture of kaempferol 3-*O*-sophoroside and kaempferol 3-*O*-sambubioside, obtained from *M. arboreus*, on the gastric concentration of interleukin-10 (IL-10, **A**) and 6 (IL-6, **B**), and enzyme activity (catalase, **C**), in Sprague Dawley rats, with ethanol-induced ulcers. Healthy = rats without gastric ulcers; L-Arg = L-arginine. ANOVA post-Tukey’s test with (n = 5, $\bar{x}$ ± ED, ^#^
*p* < 0.05 when groups are compared with the healthy group; * *p* < 0.05 compared with the Veh group).

**Figure 8 plants-11-02951-f008:**
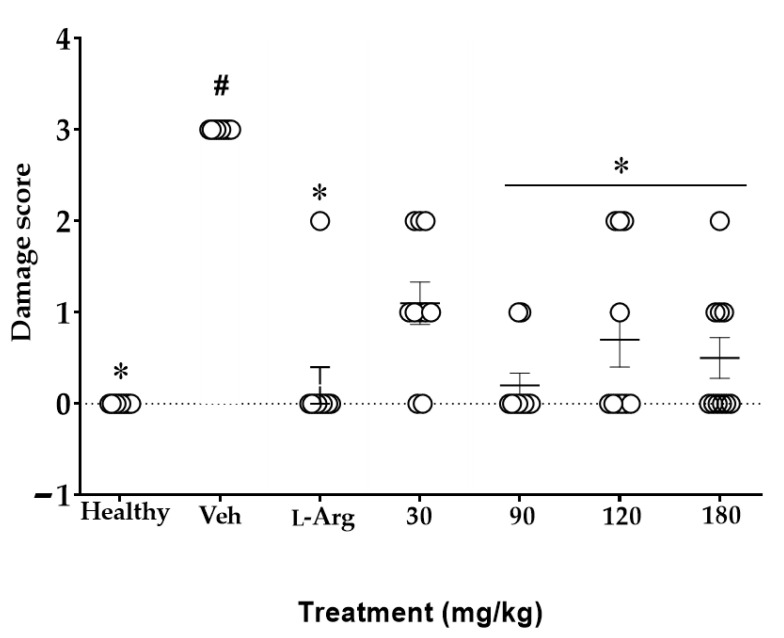
Effect of different doses of MaSS, the mixture of kaempferol 3-*O*-sophoroside and kaempferol 3-*O*-sambubioside, obtained from *M. arboreus*, on damage scores produced in the stomach of rats with ethanol. Healthy = rats without gastric lesions; L-Arg = amino acid L-arginine. Friedman was used with Dunn’s multiple comparisons *post*-test (n = 5 ± ED, ^#^
*p* < 0.05 when groups are compared with the healthy group; * *p* < 0.05 compared with the Veh group).

**Table 1 plants-11-02951-t001:** Cell count of each of the stomach strata.

Treatments (mg/kg)	Number of Cells		
	Mucosa	Muscularis Mucosae	Submucosa
Healthy	90.0 ± 5.1 *	54.6 ± 2.3 *	48.6 ± 1.1 *
Veh	30.0 ± 2.1^#^	35.6 ± 3.0	29.0 ± 2.0
L-Arg (300)	144.3 ± 7.1 *	65.3 ± 2.5 *	84.6 ± 6.5 *
MaSS			
30	47.6 ± 5	48.6 ± 7.7	69.3 ± 3.7 *
90	90.3 ± 8 *	79.0 ± 7.0 *	54.6 ± 3.5 *
120	48.6 ± 6	52.6 ± 4.1	58.6 ± 4.9 *
180	67.0 ± 5 *	53.0 ± 5.1	45.6± 1.5 *

Healthy = rat without gastric lesions, Veh = vehicle and EtOH, L-Arg = L-arginine, MaSS = mixture of kaempferol 3-*O*-sophoroside and kaempferol 3-*O*-sambubioside. ANOVA post-test Tukey (n = 5, $\bar{x}$ ± ED, ^#^
*p* < 0.05 when groups are compared with healthy group; * *p* < 0.05 compared with Veh group).

**Table 2 plants-11-02951-t002:** Sum of damage scores of variables associated to gastric lesions induced with ethanol.

Treatments (mg/kg)	Scores of Damage Variables
Healthy	0
Veh	30
L-Arg (300)	2
MaSS	
30	11
90	2
120	7
180	5

Healthy = rat without GU, Veh = vehicle and EtOH, L-Arg = L-arginine, MaSS = mixture of kaempferol 3-*O*-sophoroside and kaempferol 3-*O*-sambubioside.

**Table 3 plants-11-02951-t003:** Score values associated with damage variables in gastric ulcers induced with ethanol.

	Scores			
	0	1	2	3
Variable of Histologic Damage				
**Number of cells (300 × 300 px)**				
Mucosa	85 or more	72 to 84	47 to 71	46 or less
Muscular of Mucosae	52 or more	47 to 51	36 to 46	35 or less
Submucosa	68 or more	56 to 67	32 to 55	31 or less
**Microscopic edema (µm)**				
Mucosa	27 or less	28 to 34	35 to 48	49 or more
Muscular of Mucosae	4 or less	5	6 to 8	9 or more
Submucosa	21 or less	22 to 25	26 to 32	33 or more
**Cytokines (pg/g protein)**				
IL-6	279 or less	280 to 472	473 to 858	859 or more
IL-10	392 or more	271 to 391	29 to 270	28 or less
**Other variables**				
Relative Weight Stomach (%)	0.49 or less	0.50 to 0.57	0.58 to 0.72	0.73 or more
Gastric Ulcer (%)	11 or less	22 to 12	46 to 23	47 or more

## Data Availability

Not applicable.

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
