# Peer review of "A Mixture of Kaempferol-3-O-sambubioside and Kaempferol-3-O-sophoroside from Malvaviscus arboreus Prevents Ethanol-Induced Gastric Inflammation, Oxidative Stress, and Histologic Changes"

_plants, 2022, doi:10.3390/plants11212951_

Round 1

Reviewer 1 Report

Concerning this article, authors describe the biological activity of two flavonoids from Malvaviscus arboreus. If I understand well, this work is the continuation of the work Gastroprotective activity of kaempferol glycosides from Malvaviscus arboreus Cav., published in J of Ethnopharmacology with higher focus on biological activity. Why the authors didn't test different proportions of two compounds in order to show if there is any synergistic or antagonist activity ? It would be better to include also the results of individual flavonoids biological activity. Concerning phytochemistry, the only question I had is the same. The authors indicate the detection wavelength as 350 nm on figure and 330 nm in chapter 4.2 (line 430) in the text.

Author Response

Comments

Answer

Reviewer 1

Concerning this article, the authors describe the biological activity of two flavonoids from Malvaviscus arboreus. If I understand well, this work is the continuation of the work Gastroprotective activity of kaempferol glycosides from Malvaviscus arboreous Cav., published in J of Ethnopharmacology with higher focus on biological activity.

Why the authors didn’t test different proportions of two include also the results of individual flavonoids biological activity. 

It is a fitting observation and we appreciate it.

At this time, the main intention of this work was to use a mixture of two compounds, whose basic structure is the same and thereby reduce the number of animals to be used, since for the treatment of rats the amount of compound to be used is high.

This allows us to optimize these compounds, isolated from M. arboreous, for chemical characterization.

However, we also intend to continue working with the plant to increase its pharmacological and chemical characterization, which includes the isolation of other compounds and

evaluating them individually in models of gastric damage, which will allow us to better address the problem.

Concerning phytochemistry, phytochemistry, the only question I had is the same. The authors indicate the detection wavelength as 350 nm in the figure and 330 in chapter 4.2 (line 430) in the text.

The data was corrected in the methodology section since the correct value is at 350 nm. Marked in yellow

Section 4.2

Reviewer 2 Report

In the study titled “Kaempferol-3-O-Sambubioside and kaempferol-3-O- 2 sophoroside from Malvaviscus arboreus protect against gastric inflammation, oxidative stress, and histologic changes”, the authors studied the effects of Kaempferol-3-O-Sambubioside and kaempferol-3-O- 2 sophoroside from Malvaviscus arboreus in an acute model of gastric lesions induced by ethanol. The study is interesting and will contribute to the field. However, some modifications have to be made.

Introduction

- It is not clear in the following sentence what the authors intended to say:

“In addition, the gastroprotective agent, which promotes an increase in both the volume and pH gastric juice, decreases the weight index, the amount of mucus secreted, and ulcers in the stomach strata.”

Material and Methods section

- The ethanol-induced injury model does not mimic the features of a gastric ulcer. Therefore, I suggest that the authors change the text from “ethanol-induced gastric ulcers” to “ethanol-induced gastric lesions” throughout the text.

- The following sentence is too long and therefore, confusing: “With a light-dark cycle of 12 by 12 hours, and a temperature of 25 ± 2°C, with a constant flow of air and the feed was free of additives, hormones, anti-biotics, drugs, pesticides and pollutants, and free access to water consumption.”

- In basic research, the oral administration of substances is classically referred to as "p.o.", that is, by mouth, orally. (p.o.: Abbreviation meaning by mouth, orally / from the Latin "per os", by mouth); I suggest authors replace "op" with po or p.o.

- I suggest restructuring the sentences that describe the treatment groups.

Example: “Group 2. Vehicle, animals with ulcers and with only Tween 20 (1%) by op” / “Group 2. Vehicle, animals that received ethanol and were treated with Tween 20 (1%) (p.o.)”.

- Why did the authors use doses of 30, 90, 120, and 180 mg/kg? This explanation must be contained in the text.

- Please, change “Sacrificed” to euthanized”.

- Please specify methodologies. Phrases like “The stomachs were stored for the shortest time possible in 10% formalin until used” are very vague.

- Please specify methodologies. How was the quantification of gastric mucus done?

- Did the authors check the distribution of the data before choosing this statistical method used? For example, score data are classically described as nonparametric data, so they should be expressed as the median and interquartile range (for example) and analyzed with appropriate statistics for nonparametric data.

Results section:

- The authors do not describe the results, they only indicate p-values. Furthermore, the authors describe the data as “some treatment”. Authors need to describe the data for each group (ie, mean or median, in agreement with the characteristic of each data, error or standard deviation, or interquartile range) and compare it to the appropriate control group.

- How did the authors quantify and ensure the precise amount of gastric juice present in each stomach without ligating the pylorus? (so that stomach contents remain in the stomach).

- How do the authors explain the increase in gastric juice content for animals treated with the 90 mg/kg dose?

- Please correct the axes of the graphs. "mocus"

- The authors mention a “significant inhibition of the level of gastric ulcers when compared to Veh (* p<0.05)”. The injured area is a quantification made in pixels or in mm2 and not a measurement in levels.

- The image quality of the graphics, as well as stomachs and histology is low. Still, on the histology photos, I suggest identifying the changes with symbols, not colored arrows; and indicating the meaning of the symbols in the legend. After, restructure the histology images in the panel.

- Symbols indicating the statistical difference in graphs must be rearranged. You must use a symbol (example: # to indicate the difference between the vehicle and the disease group), and another symbol (example: *) to indicate the difference between the groups treated with the disease group.

Discussion section

- The hypothesis gets confused in the discussion. The authors bring in the introduction a review of the ulcer (general), and in the discussion, they talk about ethanol-induced lesions.

Author Response

Reviewer 2                                                                                                             Answer

In this study title “Kaempferol-3-O-Sambubioside and kaempferol-3-o-2 sophoroside from Malvaviscus arboreus protect against gastric inflammation, oxidative stress, and histologic changes”, the authors studied the effects of Kaempferol-3-O-Sambubioside and kaempferol-3-sophoroside from Malvaviscus arboreus in an acute model of gastric lesions induced by ethanol. The study is interesting and will contribute to the field. However, some modifications have to be made.

Introduction

-It is not clear in following sentence what the authors intended to say

“In addition, the gastroprotective aget, which promotes an increase in both the volume and ph gastric juice, decreases the weight index, the amount of mucus secreted, and ulcers in the stomach strata”

Thanks for the observation. The authors read the mentioned paragraph, and you are quite right, there is no connection and it is redundant with what was mentioned at the end of the introduction. In which the objective of the work is emphasized. For this reason, we decided to remove these lines.

Introduction

Lines 54-56

Material and Methods section

-The ethanol-induced injury model does not mimic the features of gastric ulcer. Therefore, I suggest that the authors change the test from

“Ethanol-induced gastric ulcers” to “ethanol-induced gastric lesion” throughout the text.

Certainly, the administration of ethanol does not completely mimic the pathophysiological mechanism associated with gastric ulcers, especially those associated with Helicobacter pylori.

However, the gastric damage observed is associated with oxidative stress, inflammation, decreased mucus, ulcer production, and changes in gastric secretion, among other factors.

The authors decided to follow the suggestion, and the changes are marked with yellow.

Also, the name of Y-axis in the figure 3 was changed

throughout the text

-The following sentence is too long and therefore, confusing:

“With a light-dark cycle of 12 by 12 hours, and a temperature of 25 ± 2°C, with a constant flow of air and the feed was free of additives, hormones, anti-biotics, drugs, pesticides and pollutants, and free access to water consumption”

The phrases were shortened, thanks for the observation.

It was marked in yellow

Section 4.3.1.

-In basic research, the oral administration of substances is classically referred to as “p.o.”, that is, by mouth, orally. (p.o.: Abreviation meaning by mouth, orally/from the Latin “per os”, by mouth); I suggest authors replace “op” with po or p.o.

The change was made, and marked in yellow

Section 4.4

-I suggest restructuring the sentences that describe the treatment groups.

Wording was changed, as suggested by reviewer. Marked in yellow

Section 4.4

-Why did the authors use doses of 30, 90, 120, and 180 mg/kg? This explanation must be contained in the text

The explanation was noted in the appropriate place, and marked in yellow

Section 4.4

-Please, change “Sacrificed” to “euthanized”

The change was made

Section 4.5

-Please specify methodologies. Phrases like “The stomachs were stored for the shortest time possible in 10% formalin until used” are very vague.

The phrase was changed to

“For only one week”

Section 4.5.6

-Please specify methodologies. How was the quantification of gastric mucus done?

Thanks for the clarification.

The methodology about this, was attached

Section 4.5.2

-Did the authors check the distribution of the data before choosing this statistical method used? For example, score data are classically describe as nonparapetric data, so they should be expressed as the median and interquartile range (for example) and analyzed with appropriate statistics for nonparametric data.

An apology in advance for this table (2), we did not actually do statistics, since we consider it a qualitative parameter. It was an error, copying the legend “ANOVA post-Dunnet test (n=6, x Ì… ±ED, *p <0.05)”, which applied to other results.

Section 2.7

Results

-The authors do not describe the results; they only indicate p-value. Furthermore, the authors describe the data as “some treatments”. Authors need to describe the data for each group (the mean or median, in agreement with the characteristic of each data, error or standard deviation, or interquartile range) and compare it to the appropriate control group.

Following this observation, all data for each graphic, have been incorporated into the text, in the results section.

We tried to describe the results and mention was made of the comparison of the Veh group with the healthy animals (#) and the comparison of the Veh group with the treatments (*)

Section, results

-How did the authors quantify ensure the precise amount of gastric juice present in each stomach without ligating the pylorus? (So that stomach contents remain in the stomach)

You’re right, we didn’t actually ligate the pylorus. What was done was to take the gastric juice with a micropipette before finishing opening the stomach. What was done in all the groups, by the same person and the same instrument.

-How do the authors explain the increase in gastric juice content for animals treated with the 90 mg/kg dose?

This variable, on the increase in the volume of gastric juice with the dose of 90 mg/kg, is probably the data that most attracts attention. However, other variables show a trend (Figures 2B, 2D, 3, 6B, 6C, 7A, table 1), which has already been observed for many phytochemicals, which is the phenomenon of hormesis.

  In which, the pharmacological response is varied (or to put it another way, biphasic), depending on the dose range.

For the results of this work, it is observed that all doses, in most of the variables, have gastroprotective activity.

Our intention is to continue with the study of these and other compounds isolated from the plant, basically phenolics of the flavonoid type, which will lead us to verify their hormetic response, in relation to their gastroprotective effects. What in the future would allow establishing the appropriate dose range for said activity.

-Please correct the axes of the graphs “mocus”

The change was made

-The authors mention a “significant inhibition of the level of gastric ulcers when compared to Veh (*p<0.05)”. The injured area is a quantification made in pixels or in mm2 and not a measurement in levels.

This observation was considered in the new version of the manuscript

-The image quality of the graphics, as well as stomachs and histology is low. Still, on the histology photos, I suggest identifying the changes with symbols, not colored arrows: and indicating the meaning of the symbols in the legend. After: restructure the histology images in the panel.

All figures, and images, were modified.

-Symbols indicating the statistical difference in graphs must be rearranged. You must use a symbol (example: # to indicate the difference between the vehicle and the disease group), and another symbol (example: *) to indicate the difference between the groups treated with the disease group

All graphics and tables were corrected

Discussion section

-The hypothesis gets confused in the discussion. The authors bring in the introduction a review of the ulcer (general), and in the discussion, they talk about ethanol-induced lesions.

Thanks a lot for your commentary.

In the introduction, with the next paragraph

“The model of gastric lesions-induced by ethanol, is widely used in rodents, for the evaluation of mechanisms associated with this pathology, and in the search of therapies based on medicinal plants. This substance damages the gastric mucosa and exposes the tissue to the actions of hydrochloric acid and pepsin, reduces blood flow and causes microvascular injuries by increasing the production of reactive oxygen species (ROS) and proinflammatory cytokines, thereby reducing levels of natural antioxidants “

We try to explain and, in particular, associate GU disease with the mentioned model. We indicate that this work is focused on trying to find new alternatives to alleviate the disease. Likewise, that a single model does not necessarily imitate the pathology on which it is sought to have a better treatment than the existing ones.

In the discussion, we continue in the same direction of thought by mentioning some of the features of the model. But without neglecting the hypothesis that "the mixture of flavonoids is capable of counteracting some of the events caused by ethanol"; however, these may be associated with the physio-pathological mechanism that generates GU, since these are multifactorial. Finally, we consider that there is an imbalance in the homeostatic mechanisms, which are also generated by ethanol.

Reviewer 3 Report

The manuscript entitled “Kaempferol-3-O-Sambubioside and kaempferol-3-O-sophoroside from Malvaviscus arboreus protect against gastric inflammation, oxidative stress, and histologic changes” describes a study where a mixture of kaempferol-3-O-Sambubioside and kaempferol-3-O-sophoroside was used to prevent animals challenged with ethanol from developing gastric ulcers. Chemical, biochemical and histological parameters of the gastric tissue and fluids were evaluated, including gastric juice volume, pH, stomach weight, ulcerated area, and gastric IL concentration, among others. The authors concluded that the mixture of both glycosides exerts a gastroprotective effect in Sprague Dawley rats, with the most effective dose being 90 mg/kg.

Although interesting, I have a few significant concerns with this manuscript, particularly with regard to (1) the relevance of the study design to finding a treatment for GU, and (2) the results of the gastric juice pH measurements. I also believe that the quality of the writing should be improved.

Major comments:

  1. I find it excessive to imply that this mixture of compounds diminishes the gastric lesions caused by ethanol in an animal model where these lesions were not yet established, particularly when the study is centered around the importance of finding a pharmaceutical treatment for gastric ulcers. If I understood correctly, the cited Morimoto method uses compound treatment 30 minutes before the ethanol challenge, which happens only once. In other words, the protocol applied in this study is designed to assess the preventive effects of both compounds against acute ethanol-induced gastric ulceration. This gastroprotective effect of MaSS is indeed mentioned in the conclusion, but I believe it should be reinforced and further contextualized in the potential application envisioned for MaSS. The way treatments are conducted should also be stated in the main text, so that readers are perfectly aware of the conditions in which MaSS is being applied as a preventive treatment.

  2. Normal gastric pH is around 1-2. In mice and rats, normal gastric content has been described to have a pH value of up to 3.9 (DOI:

10.1186/s13765-020-00536-8 and 10.1211/jpp.60.1.0008). The 8.4 value described in this manuscript seems rather off, and the authors need to justify these discrepancies - if a plausible reason indeed exists.

  1. The title of the article should be changed to “A mixture of Kaempferol-3-O-Sambubioside and kaempferol-3-O-sophoroside from Malvaviscus arboreus prevents ethanol-induced gastric inflammation, oxidative stress, and histologic changes” or an equivalent title.

Minor comments:

Line 62: If this animal model is indeed widely used, the authors should cite a few recent references to exemplify the application.

Line 69: This sentence seems to be connected to the previous one, so it is not clear why a paragraph was made. Furthermore, after the word “continuously” a brief clarification should be added, e.g. “with normal cellular activity”.

Line 100: The use of L-Arg as a positive control should be justified and properly supported by literature in its first appearance. Also, the used dosage should be clearly stated.

Figure 2: The used dosage of L-Arg should be clearly stated in the figure legend.

Lines 105-109: The sentence is poorly written and the use of the word “while” is not adequate. At the beginning of a sentence, “while” has the same meaning as “whereas”, which does not seem to be the case here. Please rephrase. 

Lines 107-108: In Fig. 2B, a statistically significant (*) decrease is marked for L-Arg, as well as 30, 120, and 180 mg/kg doses, compared to Veh. This is contradictory to what is described in this part of the text. Please double-check and make the appropriate corrections.

Line 190: The fact that there is an increase in gastric juice is not mentioned here, nor anywhere else in the text. This is particularly relevant because the 90 mg/kg dosage is highlighted in the conclusion as the most effective MaSS dose. Please discuss the 90 mg/kg dose, as well as the lack of a clear dose-response relationship (not only here, but wherever applicable).

Figure 5: Figures F-G are missing.

Table 1: “Cells number” should be replaced with “Cell count” or “Number of cells”.

Line 324: The word “arise” should be replaced with “offer” or an equivalent expression.

Line 339: The word “observing” should be replaced with “...and the results show systemic effects on the…” or a similar expression. “Observing” is not the correct option here.

Line 356: Instead of “counteracted”, please use “...a greater secretion of mucus was observed…” or an equivalent expression.

Line 365: Please rephrase or add the word “as” before “they reported increased…”.

Line 403: There is an extra space in IL-6, please remove it.

Line 444: Please remove the word “Now”.

Lines 508-532: There is no mention of the frequency of the procedure, the volume of ethanol used for ulcer induction, nor the period of time between L-Arg/MaSS and ethanol treatments. This is a crucial part of the protocol and should be presented in detail.

Lines 527-529: “this drug reduce” should be corrected to “this drug reduces”. Furthermore, this sentence was inappropriately placed in this section. It should be mentioned the first time L-Arg is presented as the positive control.

Adjustments to be made throughout the manuscript:

  1. When presenting values in the text, e.g., line 99, there should be a space after the ± sign. It should read 0.763 ± 0.136. Please correct wherever applicable.

  2. In L-Arg, as well as in carbohydrate nomenclature, D- and L- prefixes should be formatted to small capitals. Please correct wherever applicable.

  3. There is an overall inadequate use of commas throughout the manuscript. Example (lines 140-146): “Figure 3, shows that…” no comma is necessary here. The same in “induced all,...”. This is a recurring problem in all sections of the manuscript. Please revise.

  4. As exemplified in a previous comment, the word “while” is misused and excessively employed in the manuscript. This is a recurring problem, particularly in the results section. Please revise the use of the word “while” wherever applicable.

  5. Figure legends: There should be a space before and after the symbol “=”.

  6. MaSS is written as “MaSs” or “Mass” e.g. in Table 2 and on page 9.

Round 2

Reviewer 2 Report

Some modifications have been made, however, the manuscript has not yet been fully corrected.

The authors did not understand the previous question: Did the authors check the distribution of the data before choosing this statistical method used (for all data)? For example, score and % data are classically describe as nonparapetric data, so they should be expressed as the median and interquartile range (for example) and analyzed with appropriate statistics for nonparametric data. Authors cannot simply choose 1 statistical test and use the same test for all data.

Furthermore, I do not believe the measurements of gastric juice and mucus are accurate in the way they were made. Furthermore, I still feel that the introduction needs to be aligned with the discussion and hypothesis of the work.

Reviewer 3 Report

Dear all,

I am still not convinced that the authors provide enough evidence of the relevance of this study. How can these findings be seen as an important advance in the R&D of new drugs for gastric ulcers? How do the authors see the relevance of these results considering that the rats were treated before alcohol ingestion? What other studies need to be conducted to move this line of research forward? These are some of the questions I would like to see discussed in more detail in the discussion section.

I also still have some minor comments:

1. With regard to my previous comment about the Morimoto model:

“Line 62: If this animal model is indeed widely used, the authors should cite a few recent references to exemplify the application.“ 

I could not find that sentence again, despite the author’s reply: “We add cites that make use of the mentioned model”. When answering reviewer's comments, the authors should present the new sentences in the reply file and add the new location in the manuscript so that the reviewer can go there directly and quickly understand what changes were conducted.

2. Regarding my comment “Table 1: “Cells number” should be replaced with “Cell count” or “Number of cells”.”, the change was not conducted appropriately in the header.

3.  Concerning my comment “In L-Arg, as well as in carbohydrate nomenclature, D- and L- prefixes should be formatted to small capitals. Please correct wherever applicable.”, I am perfectly aware of what the “L” and “D” prefixes stand for, and they need to be formatted in small capitals. I kindly ask the authors to please carefully re-read my comment. In case of doubt, the authors can find amino acid and carbohydrate nomenclature here:

https://pubmed.ncbi.nlm.nih.gov/20251441/

4. Figure 7C is not cited in the text. Furthermore, there is no indication of statistical significance in the Figure. Please adjust.

5. Figure 8 is numbered as Figure 7. But more importantly, the Figure is very difficult to understand. Please either present the data differently or delete the Figure altogether. 

6. In kaempferol 3-O-D-sophoroside and kaempferol 3-O-D-sambubioside, the letter “O” should be in italic. In lines 62 and 542, for instance, the formatting is still incorrect.

Many thanks.

Round 3

Reviewer 2 Report

The manuscript can be accepted. 

Author Response

Thanks a lot

Reviewer 3 Report

Dear all, 

Most significant problems were addressed in the last revision round and I am now supportive of acceptance. My only comment would be that instead of "cells count" it should read "cell count". Furthermore, the correct term is "Helicobacter pylori" and not "Helicobacter pylorus" (line 452). "M. arboreous" should be in italic (line 445), as well as "in vitro" and "in vivo" (line 454).

Thank you.